# Characteristics and Epidemiology of Discharged Pneumonia Patients in South Korea Using the Korean National Hospital Discharge In-Depth Injury Survey Data from 2006 to 2017

Kyunghee Lee [1], Kyunglan Hong [2], Sunghong Kang [3] and Jieun Hwang [4,*]

1   Department of Healthcare Management, Eulji University, Seongnam-si 13135, Korea; rose1294@hanmail.net
2   Team of Medical Record, Seoul National University Hospital, Seoul 03080, Korea; hkr0775@naver.com
3   Department of Health Policy & Management, Inje University, Gimhae-si 50834, Korea; hcmkang@daum.net
4   College of Health Science, Dankook University, Cheonan-si 31116, Korea
*   Correspondence: hwang0310@dankook.ac.kr

**Abstract:** Despite the use of vaccines and various antibiotics, approximately 30% of the South Korean population is treated for pneumonia each year, and the number of deaths from pneumonia continues to increase. The present study used information on discharged patients in South Korea to investigate the number and characteristics of discharged pneumonia patients across 12 years. Using the Korean National Hospital Discharge In-Depth Injury Survey data, information on discharged patients from 2006 to 2017 were collected. The number of discharged pneumonia patients for each year and their age group was assessed, and the Charlson Comorbidity Index was used to assess the risk of comorbidities in these patients. The number of discharged pneumonia patients varied every year in South Korea. In particular, the total number of patients increased substantially in 2011, with a large increase in the number of infants and children. In addition, the number of discharged pneumonia patients increased in the elderly group compared to the other age groups. Moreover, a recent increase in the severity of comorbidities in pneumonia patients was noted. Given the continued increase in the number of elderly patients with pneumonia, chronic diseases, such as hypertension and diabetes, should be managed first in the elderly. Moreover, appropriate treatment methods should be selected based on the presence of comorbidities.

**Keywords:** pneumonia; comorbidity; epidemiology; aged; Korea

## 1. Introduction

Approximately 30% of the South Korean population is treated for pneumonia each year [1]. In particular, despite the use of vaccines and various antibiotics, mortality from pneumonia continues to increase in South Korea [1,2]. According to Statistics Korea, pneumonia was the ninth most common cause of death in South Koreans in 2009 but was the third most common cause in 2018. The high mortality associated with pneumonia is especially noticeable in male and elderly patients, which can be explained by the increase in patients with chronic diseases due to the aging of the population [3].

Pneumonia is a severe respiratory disease characterized by inflammation in the lungs. In pneumonia, bacteria or viruses cause inflammatory reactions in the lung parenchyma below the level of bronchioles [4]. Depending on how it is acquired, pneumonia is classified as community-acquired pneumonia (CAP), hospital-acquired pneumonia, and healthcare-associated pneumonia [4]. Such classification helps determine therapeutic approaches and prognosis for individual patients by providing information on the distribution and antibiotic resistance of pathogens and comorbidities in the patients [5].

Pneumonia is more common in patients with other underlying conditions, such as obstructive lung disease, diabetes, heart failure, and chronic liver disease. Moreover, pneumonia is a major complication and a disease that results in high mortality when

accompanied by other serious diseases, such as coronavirus disease 2019, acquired immunodeficiency syndrome, and cancer. Therefore, it is necessary to assess pneumonia along with comorbidities [6]. Above all, since comorbidities may change clinical presentation and determine future treatment options [7], research investigating pneumonia and comorbidities is of utmost importance.

However, pneumonia research thus far has focused on identifying risk factors; for instance, studies have investigated risk factors or cellular immune mechanisms associated with CAP and outbreaks [8,9] or analyzed the type of and risk factors for pneumonia in infants and children [10]. Therefore, the present study aimed to assess the number of discharged pneumonia patients in South Korea and their comorbidities using 12 years of data. Understanding the epidemiology of pneumonia in South Korea is important, as it can serve as important evidence in establishing international clinical guidelines and treatment strategies.

## 2. Materials and Methods

### 2.1. Study Data

This study used the Korean National Hospital Discharge In-Depth Injury Survey data to collect information of discharged patients from 2006 to 2017. The data are collected by the Korea Disease Control and Prevention Agency to establish cost-effective national public health policies through an understanding of time-series trends in the number and characteristics of discharged patients (IRB no. EU21-020) [11,12].

The target population for the Korean National Hospital Discharge In-Depth Injury Survey consists of all patients discharged from general hospitals across the country, and approximately 9% of the total discharged patients were discharged from 170 hospitals selected according to hospital size, and survey methods comprised the sample for the survey. The survey was based on medical records and included data regarding patients' demographics, hospitalization, diseases, treatment, and external causes of injuries.

This study defined pneumonia cases as those with J12–J18 as the main condition according to the ICD-7th edition [13]. Since the Korean National Hospital Discharge In-Depth Injury Survey includes up to 20 sub-diagnoses, each pneumonia patient's sub-diagnosis was defined as comorbidity [14].

### 2.2. Study Model and Analysis

For time-series trends in male and female discharged pneumonia patients each year, the number of patients diagnosed each year from 2006 to 2017 and the total number of patients in South Korea were assessed separately. Since the Korean National Hospital Discharge In-Depth Injury Survey uses a two-stage stratified cluster sampling method, complex sampling methods can be applied to calculate the weighted total number of discharged patients and the weighted discharge rate.

To understand the characteristics of discharged pneumonia patients, their sex, age group, admission route, insurance type, treatment outcome, number of hospital beds, and length of stay were analyzed according to age (0–6 years old: infants and children; 7–18 years old: adolescents; 19–64 years old: adults; and ≥65 years: senior) and year for frequency analysis. For frequency, the number of discharged pneumonia patients in each year and the weighted percentage calculated based on the number of all patients were presented. Differences in distribution according to age and year were tested for statistical significance using the chi-square analysis.

The Charlson Comorbidity Index (CCI) was used to assess the risk of comorbidities in discharged pneumonia patients. The CCI calculates the score for comorbidities according to the relative risks of 19 major diseases, including ischemic heart disease, diabetes, and hypertension [15]. For CCI, codes converted to the algorithm from the ICD-10 codes were used for the analysis. The number of patients and the weighted percentage for each index were used to assess the risk of comorbidities. Differences according to age and year were tested for statistical significance using the chi-square analysis.

Statistical analyses were performed using SAS version 9.4 (Cary, NC, USA), and statistical significance was set at *p* < 0.05.

## 3. Results

*3.1. Number of Discharged Pneumonia Patients in South Korea (2006–2017)*

Figure 1 shows the number of discharged pneumonia patients each year. In 2006, the total number of discharged pneumonia patients was 6421: 3958 children, 490 adolescents, 909 adults, and 1064 seniors. In 2017, the total number of patients was 10,648: 4668 children, 349 adolescents, 1805 adults, and 3826 seniors. Compared to 2006, the total number of discharged pneumonia patients increased by 66%, and the increase was observed across all age groups (18% in children, 99% in adults, and 260% in seniors) except in adolescents (29% decrease). The number of discharged patients with pneumonia fluctuated each year. The number was lowest in 2008 at 5472 patients and highest in 2011 at 12,994 patients. The number of discharged pneumonia patients decreased after 2011 but started increasing again in 2014.

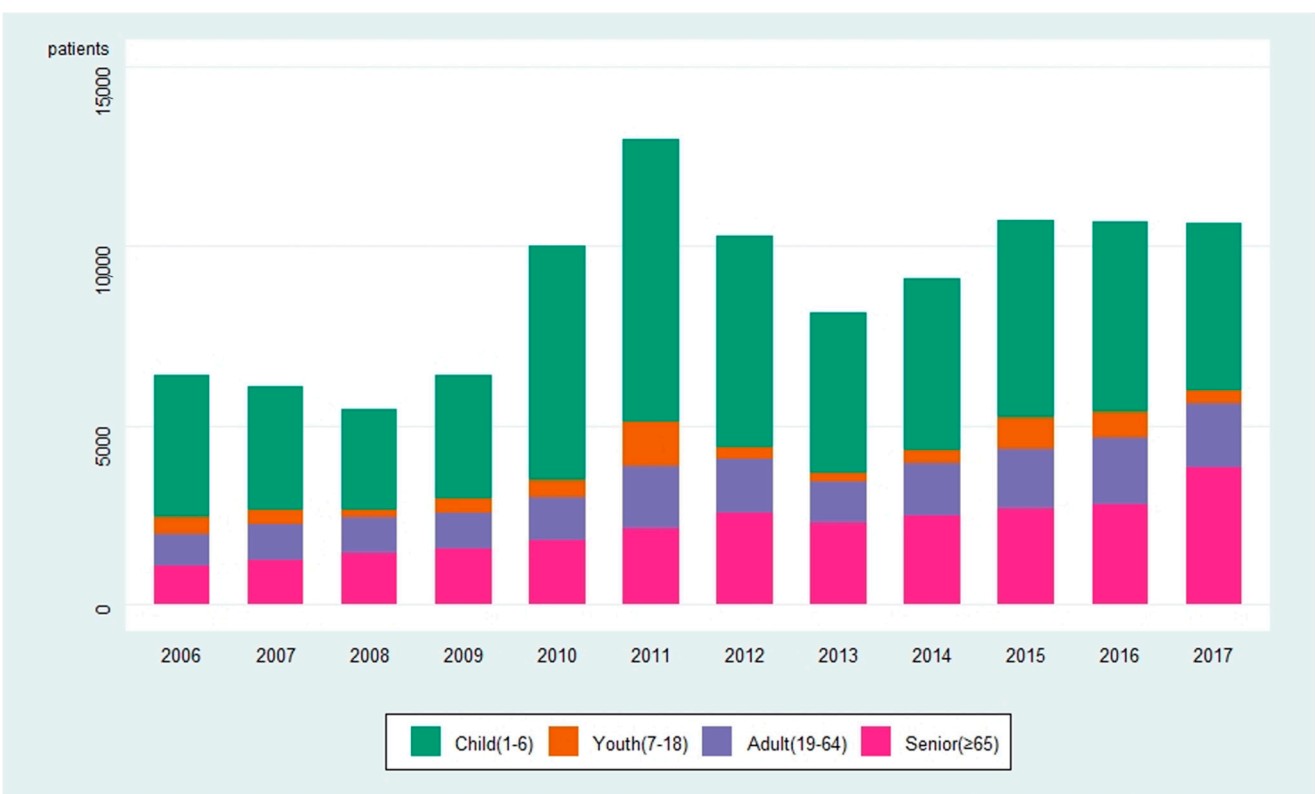

**Figure 1.** The number of discharged pneumonia patients in each year and age group (2006–2017).

In terms of the weighted percentage of discharged pneumonia patients compared to the total number of discharged patients in South Korea each year, a decreasing trend was noted in children since 2011, but in seniors, this trend was increasing. The number of adolescents and adults fluctuated and showed a decreasing trend since 2015 (Figure 2).

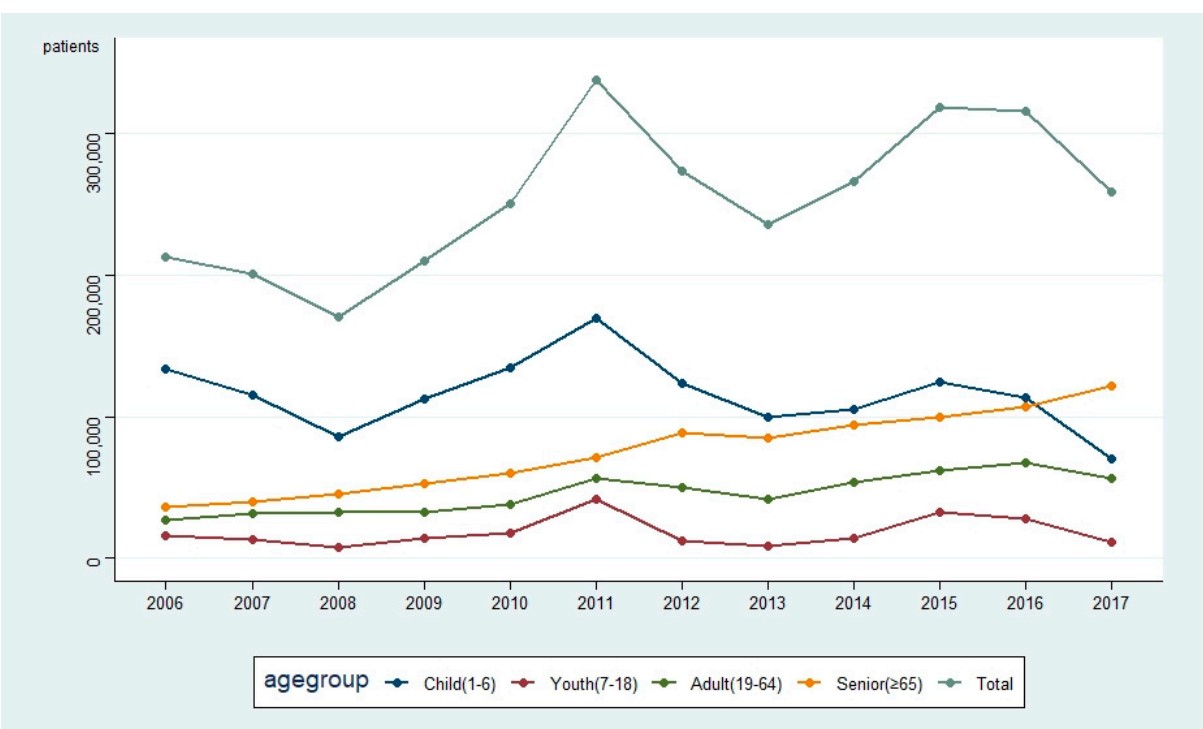

**Figure 2.** Weighted number of discharged pneumonia patients in each year and age group (2006–2017).

As of 2017, the number of pneumonia patients per 10,000 persons by region was as shown in Figure 3. The region with the highest number of discharged pneumonia patients per 10,000 persons was Jeju, followed in order by Ulsan, Jeollabuk-do, Gyeongsangnam-do, and Chungcheongnam-do.

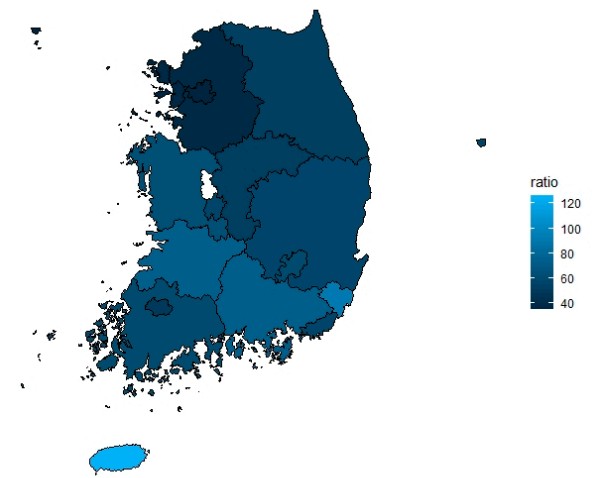

**Figure 3.** Weighted number of discharged pneumonia patients by regional population in South Korea (2017) (unit: per 10,000 person).

*3.2. Demographic Characteristics and Admission Information of Discharged Pneumonia Patients*

3.2.1. Age Group

Table 1 shows the demographic characteristics and admission information of pneumonia patients in each age group.

**Table 1.** Demographic characteristics and admission information of discharged pneumonia patients in South Korea in each age group (2006–2017).

| | Child (0–6 Years) | | | Youth (7–18 Years) | | | Adult (19–64 Years) | | | Senior (≥65 Years) | | | *p*-Value * |
|---|---|---|---|---|---|---|---|---|---|---|---|---|---|
| | *n* | Weighted *n* | Weighted % | *n* | Weighted *n* | Weighted % | *n* | Weighted *n* | Weighted % | *n* | Weighted *n* | Weighted % | |
| Sex | | | | | | | | | | | | | <0.0001 |
| Men | 31,880 | 746,135 | 53.8 | 3113 | 112,526 | 52.1 | 8063 | 274,562 | 50.2 | 14,446 | 483,778 | 53.7 | |
| Women | 26,746 | 639,991 | 46.2 | 2897 | 103,562 | 47.9 | 8351 | 272,749 | 49.8 | 11,463 | 416,522 | 46.3 | |
| Admission route | | | | | | | | | | | | | <0.0001 |
| Emergency department | 16,850 | 316,352 | 22.8 | 1862 | 55,503 | 25.7 | 7449 | 215,162 | 39.3 | 15,512 | 496,181 | 55.1 | |
| Outpatient department | 41,749 | 1,069,211 | 77.1 | 4148 | 160,585 | 74.3 | 8957 | 331,983 | 60.7 | 10,371 | 403,529 | 44.8 | |
| Others | 27 | 562 | 0.1 | 0 | 0 | 0.0 | 8 | 166 | 0.0 | 26 | 591 | 0.1 | |
| Insurance type | | | | | | | | | | | | | <0.0001 |
| NHI | 56,808 | 1,338,553 | 96.6 | 5541 | 198,497 | 91.9 | 14,708 | 489,121 | 89.4 | 22,317 | 765,991 | 85.1 | |
| Medicaid | 1616 | 43,587 | 3.1 | 452 | 17,145 | 7.9 | 1528 | 52,796 | 9.6 | 3324 | 125,398 | 13.9 | |
| Others | 202 | 3986 | 0.3 | 17 | 446 | 0.2 | 178 | 5394 | 1.0 | 268 | 8911 | 1.0 | |
| Treatment outcome | | | | | | | | | | | | | <0.0001 |
| Improved | 58,226 | 1,375,197 | 99.2 | 5940 | 213,230 | 98.7 | 15,358 | 512,130 | 93.6 | 21,330 | 740,978 | 82.3 | |
| Not improved | 287 | 8277 | 0.6 | 52 | 2110 | 1.0 | 402 | 16,559 | 3.0 | 1088 | 45,106 | 5.0 | |
| Diagnosis only | 69 | 1827 | 0.1 | 9 | 383 | 0.2 | 116 | 3753 | 0.7 | 266 | 8581 | 1.0 | |
| Others | 17 | 325 | 0.0 | 1 | 53 | 0.0 | 19 | 600 | 0.1 | 82 | 2274 | 0.3 | |
| Death | 27 | 500 | 0.0 | 8 | 312 | 0.1 | 519 | 14,269 | 2.6 | 3143 | 103,363 | 11.5 | |
| Number of hospital beds | | | | | | | | | | | | | <0.0001 |
| 100–299 | 13,410 | 543,068 | 39.2 | 2024 | 105,116 | 48.6 | 5942 | 294,241 | 53.8 | 9923 | 496,143 | 55.1 | |
| 300–499 | 10,310 | 365,999 | 26.4 | 962 | 50,281 | 23.3 | 2097 | 95,827 | 17.5 | 3159 | 149,790 | 16.6 | |
| 500–999 | 29,168 | 420,146 | 30.3 | 2551 | 53,302 | 24.7 | 6609 | 130,354 | 23.8 | 10,284 | 213,170 | 23.7 | |
| ≥1000 | 5738 | 56,913 | 4.1 | 473 | 7388 | 3.4 | 1766 | 26,888 | 4.9 | 2543 | 41,198 | 4.6 | |
| Length of stay | | | | | | | | | | | | | <0.0001 |
| ≤3 | 17,795 | 377,415 | 27.2 | 1319 | 44,878 | 20.8 | 2430 | 81,275 | 14.8 | 3128 | 112,487 | 12.5 | |
| 4–7 | 34,170 | 821,646 | 59.9 | 3465 | 123,957 | 57.4 | 6793 | 225,053 | 41.1 | 7242 | 251,258 | 27.9 | |
| 8–13 | 6108 | 171,754 | 12.4 | 1078 | 41,654 | 19.3 | 4745 | 162,041 | 29.6 | 8483 | 292,846 | 32.5 | |
| 14–20 | 384 | 10,895 | 0.8 | 101 | 3950 | 1.8 | 1291 | 43,956 | 8.0 | 3316 | 116,991 | 13.0 | |
| ≥21 | 169 | 4416 | 0.3 | 47 | 1649 | 0.8 | 1155 | 34,986 | 6.4 | 3740 | 126,719 | 14.1 | |

* Chi-square test.

In all age groups, there were more male patients than female patients ($p < 0.001$). In terms of the route of admission, outpatient admission was most common in children, adolescents, and adults each year, and the percentage of admission through emergency rooms was higher in seniors than in other age groups ($p < 0.001$). For insurance type, the percentage of national health insurance was higher in all age groups, but the percentage of Medicaid increased with increasing age ($p < 0.001$). In terms of the number of beds in hospitals from which the patients were discharged, the percentage of hospitals with 100–299 beds was the highest in all age groups, followed by those with 500–999 beds, 300–499 beds, and ≥1000 beds ($p < 0.001$). For treatment outcome, improvement, which corresponds to complete recovery or symptomatic improvement, was the most common outcome in all age groups. However, the percentage of patients with no improvement and death increased with increasing age ($p < 0.001$). In terms of the length of stay, 4–7 days was the most common in all age groups except for seniors. Following this, three days or less was common in children and adolescents, and 8–13 days was common in adults ($p < 0.001$). In contrast, in seniors, 8–13 days was the most common, followed by 4–8 days, ≥21 days or more, 14–20 days, and ≤3 days.

### 3.2.2. Yearly Changes

Table 2 shows the demographic characteristics and admission information of the discharged pneumonia patients in each year.

The difference between the percentage of male and female patients was the highest in 2010 (male weighted percentage: 4.5%; female weighted percentage: 3.7%), with a difference of 0.7%. In 2017, the difference decreased to 0.3% (male weighted percentage: 4.4%; female weighted percentage: 4.1%) ($p < 0.001$). In terms of the route of admission, the number of admissions through emergency rooms and outpatient admission was the highest in 2011 and the lowest in 2008 ($p < 0.001$). For insurance, the percentage of Medicaid showed an increasing trend recently ($p < 0.001$). In terms of treatment outcome, an increasing trend in improvement (complete recovery or symptomatic improvement) was noted along with an increasing trend in deaths ($p < 0.001$). The number of beds at hospitals from which the patients were discharged did not differ significantly across the years ($p = 0.9425$). For length of stay, admission for ≤ 3 days decreased since 2006, but admission for ≥8 days showed an increasing trend ($p < 0.001$).

### 3.2.3. Risk of Comorbidities

Table 3 shows the distribution of comorbidities according to CCI. In children and adolescents, 0 was the most common CCI. In adults, CCI 1 was the most common. In seniors, there was no index of 0 or 1, and CCI 2 was approximately three times more common than CCI 3.

Table 4 shows the number and percentage of patients according to the comorbidity index calculated using the CCI. For yearly trends in CCI, the percentage of CCI 0 decreased, and the percentage of CCI 1 and above showed an increasing trend ($p < 0.001$). In particular, the increase was most notable in the percentage of CCI 2.

**Table 2.** Demographic characteristics and admission information of discharged pneumonia patients in South Korea in each year (2006–2017).

| | 2006 | | 2007 | | 2008 | | 2009 | | 2010 | | 2011 | | 2012 | | 2013 | | 2014 | | 2015 | | 2016 | | 2017 | | *p*-Value * |
|---|---|---|---|---|---|---|---|---|---|---|---|---|---|---|---|---|---|---|---|---|---|---|---|---|---|
| | *n* | w% | *n* | w% | *n* | w% | *n* | w% | *n* | w% | *n* | w % | *n* | w% | *n* | w% | *n* | w% | *n* | w % | *n* | w % | *n* | w% | |
| Sex | | | | | | | | | | | | | | | | | | | | | | | | | 0.0004 |
| Men | 3495 | 3.8 | 3303 | 3.5 | 3023 | 3.0 | 3535 | 3.8 | 5471 | 4.5 | 6817 | 5.8 | 5578 | 4.8 | 4434 | 4.1 | 4921 | 4.6 | 5661 | 5.4 | 5559 | 5.3 | 5705 | 4.4 | |
| Women | 2926 | 3.2 | 2810 | 3.1 | 2449 | 2.6 | 2864 | 3.1 | 4526 | 3.7 | 6177 | 5.3 | 4705 | 4.2 | 3731 | 3.6 | 4161 | 4.1 | 5059 | 5.0 | 5106 | 5.0 | 4943 | 4.1 | |
| Age group | | | | | | | | | | | | | | | | | | | | | | | | | <0.0001 |
| Child | 3958 | 4.4 | 3445 | 3.8 | 2805 | 2.8 | 3439 | 3.7 | 6512 | 4.4 | 7883 | 5.6 | 5885 | 4.0 | 4489 | 3.3 | 4762 | 3.4 | 5489 | 4.1 | 5291 | 3.7 | 4668 | 2.3 | |
| Youth | 490 | 0.5 | 402 | 0.4 | 213 | 0.2 | 374 | 0.4 | 489 | 0.6 | 1237 | 1.4 | 335 | 0.4 | 212 | 0.3 | 342 | 0.4 | 863 | 1.1 | 704 | 0.9 | 349 | 0.4 | |
| Adult | 909 | 0.9 | 1026 | 1.0 | 1018 | 1.1 | 1025 | 1.0 | 1210 | 1.3 | 1754 | 1.8 | 1477 | 1.6 | 1179 | 1.4 | 1475 | 1.8 | 1679 | 2.0 | 1857 | 2.2 | 1805 | 1.8 | |
| Senior | 1064 | 1.2 | 1240 | 1.3 | 1436 | 1.5 | 1561 | 1.7 | 1786 | 2.0 | 2120 | 2.3 | 2586 | 2.9 | 2285 | 2.8 | 2503 | 3.1 | 2689 | 3.3 | 2813 | 3.5 | 3826 | 4.0 | |
| Admission route | | | | | | | | | | | | | | | | | | | | | | | | | 0.0003 |
| Emergency department | 2259 | 2.1 | 2289 | 2.1 | 2196 | 2.0 | 2596 | 2.4 | 3915 | 3.0 | 4626 | 3.6 | 3856 | 3.2 | 3134 | 2.9 | 3485 | 3.2 | 3985 | 3.6 | 4482 | 4.0 | 4850 | 3.5 | |
| Outpatient department | 4157 | 4.9 | 3822 | 4.5 | 3274 | 3.6 | 3799 | 4.5 | 6078 | 5.2 | 8363 | 7.5 | 6412 | 5.7 | 5029 | 4.9 | 5596 | 5.5 | 6731 | 6.8 | 6178 | 6.4 | 5786 | 5.0 | |
| Others | 5 | 0.0 | 2 | 0.0 | 2 | 0.0 | 4 | 0.0 | 4 | 0.0 | 5 | 0.0 | 15 | 0.0 | 2 | 0.0 | 1 | 0.0 | 4 | 0.0 | 5 | 0.0 | 12 | 0.0 | |
| Insurance type | | | | | | | | | | | | | | | | | | | | | | | | | 0.0062 |
| NHI | 5817 | 6.3 | 5518 | 6.0 | 4957 | 5.0 | 5878 | 6.3 | 9384 | 7.5 | 12,280 | 10.3 | 9610 | 8.2 | 7603 | 7.0 | 8503 | 8.0 | 10,077 | 9.7 | 9927 | 9.5 | 9820 | 7.6 | |
| Medicaid | 575 | 0.6 | 507 | 0.6 | 485 | 0.5 | 487 | 0.6 | 544 | 0.6 | 645 | 0.7 | 610 | 0.7 | 514 | 0.6 | 523 | 0.6 | 587 | 0.7 | 685 | 0.8 | 758 | 0.8 | |
| Others | 29 | 0.0 | 88 | 0.1 | 30 | 0.0 | 34 | 0.0 | 69 | 0.1 | 69 | 0.1 | 63 | 0.1 | 48 | 0.0 | 56 | 0.1 | 56 | 0.1 | 53 | 0.0 | 70 | 0.1 | |
| Treatment outcome | | | | | | | | | | | | | | | | | | | | | | | | | <0.0001 |
| Improved | 6129 | 6.6 | 5813 | 6.3 | 5133 | 5.2 | 6020 | 6.5 | 9532 | 7.7 | 12,428 | 10.5 | 9679 | 8.3 | 7616 | 7.1 | 8530 | 8.1 | 10,068 | 9.7 | 10,062 | 9.6 | 9844 | 7.7 | |
| Not improved | 115 | 0.2 | 85 | 0.1 | 98 | 0.1 | 125 | 0.2 | 130 | 0.2 | 184 | 0.2 | 168 | 0.2 | 151 | 0.2 | 144 | 0.2 | 198 | 0.3 | 191 | 0.3 | 240 | 0.3 | |
| Diagnosis only | 26 | 0.0 | 31 | 0.0 | 22 | 0.0 | 29 | 0.0 | 37 | 0.0 | 49 | 0.0 | 43 | 0.0 | 40 | 0.0 | 50 | 0.1 | 52 | 0.1 | 43 | 0.0 | 38 | 0.0 | |
| Others | 9 | 0.0 | 9 | 0.0 | 17 | 0.0 | 7 | 0.0 | 9 | 0.0 | 13 | 0.0 | 12 | 0.0 | 5 | 0.0 | 10 | 0.0 | 10 | 0.0 | 8 | 0.0 | 10 | 0.0 | |
| Death | 142 | 0.1 | 175 | 0.2 | 202 | 0.2 | 218 | 0.2 | 289 | 0.3 | 320 | 0.3 | 381 | 0.4 | 353 | 0.4 | 348 | 0.4 | 392 | 0.4 | 361 | 0.4 | 516 | 0.5 | |
| Number of hospital beds | | | | | | | | | | | | | | | | | | | | | | | | | 0.9425 |
| 100–299 | 1298 | 2.7 | 1521 | 2.7 | 1446 | 2.4 | 1645 | 3.0 | 2602 | 3.4 | 4061 | 5.2 | 3470 | 4.4 | 2668 | 3.8 | 3142 | 4.6 | 3451 | 5.1 | 3330 | 5.1 | 2665 | 4.7 | |
| 300–499 | 924 | 1.3 | 1013 | 1.7 | 747 | 1.2 | 1002 | 1.6 | 1821 | 2.2 | 1947 | 2.5 | 1510 | 2.0 | 1145 | 1.6 | 1329 | 1.8 | 1869 | 2.4 | 1615 | 2.2 | 1606 | 1.1 | |
| 500–999 | 3448 | 2.6 | 2893 | 1.9 | 2692 | 1.8 | 3136 | 2.1 | 4528 | 2.2 | 5905 | 2.9 | 4366 | 2.2 | 3636 | 2.0 | 3769 | 2.0 | 4435 | 2.5 | 4807 | 2.6 | 4997 | 2.1 | |
| ≥1000 | 751 | 0.3 | 686 | 0.3 | 587 | 0.3 | 616 | 0.3 | 1046 | 0.4 | 1081 | 0.4 | 937 | 0.4 | 716 | 0.3 | 842 | 0.3 | 965 | 0.4 | 913 | 0.4 | 1380 | 0.6 | |
| Length of stay | | | | | | | | | | | | | | | | | | | | | | | | | <0.0001 |
| ≤3 | 1208 | 1.2 | 1199 | 1.2 | 1034 | 1.0 | 1312 | 1.4 | 2266 | 1.7 | 3108 | 2.3 | 2436 | 1.9 | 2116 | 1.7 | 2348 | 1.9 | 2561 | 2.2 | 2625 | 2.2 | 2459 | 1.5 | |
| 4–7 | 3320 | 3.5 | 3100 | 3.3 | 2610 | 2.7 | 3129 | 3.4 | 5263 | 4.1 | 6705 | 5.5 | 5082 | 4.2 | 3681 | 3.3 | 4121 | 3.8 | 5115 | 4.8 | 4987 | 4.6 | 4557 | 3.4 | |
| 8–13 | 1360 | 1.7 | 1225 | 1.4 | 1220 | 1.3 | 1296 | 1.4 | 1722 | 1.6 | 2289 | 2.3 | 1833 | 1.9 | 1445 | 1.6 | 1642 | 1.9 | 2066 | 2.3 | 2023 | 2.3 | 2293 | 2.2 | |
| 14–20 | 277 | 0.3 | 297 | 0.3 | 316 | 0.3 | 317 | 0.3 | 344 | 0.3 | 470 | 0.5 | 463 | 0.5 | 442 | 0.5 | 492 | 0.6 | 504 | 0.6 | 515 | 0.6 | 655 | 0.7 | |
| ≥21 | 256 | 0.3 | 292 | 0.3 | 292 | 0.3 | 345 | 0.3 | 402 | 0.4 | 422 | 0.4 | 469 | 0.5 | 481 | 0.6 | 479 | 0.5 | 474 | 0.5 | 515 | 0.6 | 684 | 0.7 | |

* Chi-square test. * w %: weighted %.

**Table 3.** The number of discharged pneumonia patients in South Korea according to comorbidity index calculated using Charlson Comorbidity index.

| Index | Child (0–6 Years) (*n* = 58,632) | | Youth (7–18 Years) (*n* = 6010) | | Adult (19–64 Years) (*n* = 16,414) | | Senior (≥65 Years) (*n* = 25,909) | | Total (*n* = 106,965) | | *p*-Value * |
|---|---|---|---|---|---|---|---|---|---|---|---|
| | *n* | w % | *n* | w % | *n* | w % | *n* | w % | *n* | w % | |
| 0 | 58,492 | 45.3 | 5960 | 7.0 | 4698 | 4.7 | 0 | 0.0 | 69,150 | 57.1 | <0.0001 |
| 1 | 131 | 0.1 | 49 | 0.1 | 7471 | 8.6 | 0 | 0.0 | 7651 | 8.8 | |
| 2 | 1 | 0.0 | 1 | 0.0 | 3926 | 4.3 | 19,502 | 22.6 | 23,430 | 26.9 | |
| 3+ | 2 | 0.0 | 0 | 0.0 | 319 | 0.3 | 6407 | 7.0 | 6728 | 7.3 | |

* Chi-square test. * w %: weighted %.

**Table 4.** Comorbidity index of discharged pneumonia patients in each year.

| Index | 2006 | | 2007 | | 2008 | | 2009 | | 2010 | | 2011 | | 2012 | | 2013 | | 2014 | | 2015 | | 2016 | | 2017 | | *p*-Value * |
|---|---|---|---|---|---|---|---|---|---|---|---|---|---|---|---|---|---|---|---|---|---|---|---|---|---|
| | *n* | w % | *n* | w % | *n* | w % | *n* | w % | *n* | w % | *n* | w % | *n* | w % | *n* | w % | *n* | w % | *n* | w % | *n* | w % | *n* | w % | |
| 0 | 4717 | 5.1 | 4164 | 4.5 | 3250 | 3.3 | 4082 | 4.4 | 7352 | 5.3 | 9781 | 7.6 | 6551 | 4.8 | 4964 | 3.9 | 5439 | 4.2 | 6870 | 5.7 | 6577 | 5.2 | 5403 | 3.0 | |
| 1 | 420 | 0.4 | 455 | 0.5 | 479 | 0.5 | 459 | 0.5 | 538 | 0.6 | 742 | 0.8 | 756 | 0.9 | 582 | 0.7 | 746 | 0.9 | 768 | 1.0 | 859 | 1.1 | 847 | 0.9 | <0.001 |
| 2 | 1012 | 1.1 | 1167 | 1.2 | 1383 | 1.4 | 1398 | 1.5 | 1608 | 1.8 | 1966 | 2.2 | 2307 | 2.6 | 2010 | 2.4 | 2254 | 2.8 | 2361 | 2.9 | 2528 | 3.2 | 3436 | 3.6 | |
| 3+ | 272 | 0.3 | 327 | 0.3 | 360 | 0.3 | 460 | 0.5 | 499 | 0.5 | 505 | 0.5 | 669 | 0.7 | 609 | 0.7 | 643 | 0.8 | 721 | 0.8 | 701 | 0.8 | 962 | 1.0 | |

* Chi-square test. * w %: weighted %.

## 4. Discussion

Given the increasing mortality associated with pneumonia in South Korea, this study used 12 years of information on discharged pneumonia patients to analyze yearly trends South Korea. We found that the number of discharged pneumonia patients fluctuated from year to year. In particular, the total number of patients increased substantially in 2011, with a large increase in the number of infants and children. Moreover, the number of discharged pneumonia patients was on the rise in the elderly group compared to the other age groups. The severity of comorbidities showed an increasing due to increases in elderly patients with pneumonia.

In terms of changes in the number of discharged pneumonia patients, the total number of pneumonia patients seems to have increased substantially in 2011 due to the outbreak from the epidemiological cycle of *Mycoplasma pneumoniae*. In addition, pneumonia caused by *Mycoplasma pneumoniae* was designated as a legally notifiable disease in 2011 [16]. *Mycoplasma pneumoniae* infection causes respiratory illnesses, mostly in toddlers and school-aged children [17,18]. In particular, the incidence of *Mycoplasma pneumoniae* infections tends to increase with a cycle of 3–4 years. In South Korea, the incidence increased in 2007 (positivity 9.1%), 2011 (positivity 14.1%), and 2015 (positivity 11.0%) [19]. Considering that the infection was designated as a legally notifiable infection in 2011, these changes would have influenced the increase in the total number of pneumonia patients in 2011.

Of the various acute respiratory infections, *Mycoplasma pneumoniae* infections are surveyed through sentinel surveillance (clinical surveillance). Sentinel surveillance is used to continuously monitor the occurrence of acute respiratory infections in South Korea, to inform public and medical professionals about the size and signs of outbreaks, and to prepare for and promptly respond to outbreaks [20]. However, since complete enumeration is difficult, only sampling institutions that satisfy the criteria, such as hospitals with ≥200 beds and public hospitals, are required to report to respective public health center directors within seven days of confirming cases. As a result, changes in the number of institutions participating in sentinel surveillance may lead to changes in the report rate and, consequently, changes in the number of patients [21]. Therefore, the change in reporting policy in 2011 could have led to an increase in the number of patients, and this should be noted when interpreting the actual number of discharged pneumonia patients.

In terms of changes in pneumonia patients by age, it is noteworthy that the number of senior patients aged ≥65 years showed an increasing trend. This coincides with international trends, where an increase in pneumonia cases is noted in elderly patients. In the Netherlands, the admission rate due to pneumonia increased by 5% each year in 2006–2007 compared to 1999–2000, and the increase was more noticeable in elderly patients aged ≥65 years than in other age groups [22]. In England, the admission rate from pneumonia showed an increasing trend each year between 1997 and 2005, with a particularly noticeable increase in the elderly population [23]. In Denmark, the admission rate from pneumonia continuously increased between 1997 and 2011, and hospitalizations with secondary pneumonia more than doubled in seniors [24]. The increase in pneumonia in the elderly can be explained by the aging of the population. Considering that South Korea's society is aging rapidly at an unprecedented rate, the number of elderly patients with pneumonia is expected to increase even further [22].

A noteworthy aspect of the increasing trend in senior patients with pneumonia is the fact that older patients with pneumonia have more comorbidities. This increases the difficulty in therapeutic management. In fact, children and adults stayed at hospitals for 3–7 days on average, whereas senior patients with many comorbidities stayed for 8–21 days on average, indicating that complications and difficulties in therapeutic management resulting from comorbidities influenced the length of hospital stay [25].

The total number of deaths due to pneumonia was 3697. Of these, 3143 were elderly patients with comorbidities, accounting for 85.0% of the total. This finding indicates that comorbidities in senior pneumonia patients are associated with disease severity and mortality and can be interpreted in line with previous research findings [26]. In other words, comorbid

chronic conditions can interfere with treatment and create many barriers to self-management, including polypharmacy, regulation of various drugs, and financial burden.

In terms of the risk of comorbidities in pneumonia patients, there was no CCI score of 0 or 1 in seniors, but CCI scores of 0 and 1 were common in children and adolescents. In contrast, CCI 3 and 4 were more common in adults and seniors, indicating more comorbidities, and the findings confirmed that the risk of comorbidities increased with increasing age. This is because of the increasing presence of comorbidities in adults and seniors, and comorbidities can be interpreted as factors that increase the severity of pneumonia and cause difficulties in treatment [26]. In fact, four out of five elderly individuals in South Korea have chronic diseases, and almost half of this population live with three or more chronic diseases [27]. Given the increasing mortality of pneumonia, management of chronic diseases, such as hypertension and diabetes, should precede pneumonia treatment, and appropriate treatment modalities should be chosen based on the presence of comorbidities [28].

Diabetes, one of the three most common lifestyle diseases, causes diminished immune function and thus increases the incidence of certain infections. In particular, the incidence of pneumonia is approximately two-fold higher in patients with diabetes [29]. In fact, trends in the increase in patients with diabetes are known to follow the trends in the increase of respiratory infections, including pneumonia and tuberculosis [30]. Moreover, hypertension, which is known as a degenerative vascular disease, is a common chronic disease that is seen in many age groups due to westernization of lifestyle, leading to increases in rates of obesity and being overweight [31]. In addition to cardiovascular diseases, hypertension could increase the risk of diabetes, and this demonstrates a high level of correlation between chronic diseases [32]. In short, the present study confirmed the importance of managing chronic diseases in the elderly.

The present study's findings, which provide trends in the incidence of pneumonia and comorbidities in pneumonia patients, are expected to serve as important evidence in establishing clinical guidelines and treatment strategies for pneumonia. In particular, as data on discharged patients collected from 170 hospitals by the Korea Disease Control and Prevention Agency for the establishment of public health policies was used, the information presented is accurate. However, a limitation of the present study is that coding errors in the diagnostic codes used in the study data were not reviewed. This can also be seen as a limitation in monitoring the national raw data on disease statistics. Moreover, since the study defined discharged pneumonia patients using only the main condition diagnostic code, the number of pneumonia patients might have been underestimated. In addition, even though patients were discharged with pneumonia as the main diagnosis, it was difficult to confirm whether the pneumonia was secondary to another condition. Moreover, despite the fact that pneumonia is associated with personal health behavior, such as smoking, drinking, and living environment [33], these factors were not considered due to the limitation in the data used in this study. Therefore, future studies should observe the pattern of pneumonia with consideration at individual, hospital, and regional levels.

This study used information on discharged patients in South Korea to investigate the yearly trends in discharged pneumonia patients between 2006 and 2017. We found that the number of discharged pneumonia patients changed each year and that the number increased by 66% in 2017 compared to 2006. In particular, the increase was most notable in 2011, which may have been due to the cycle of *Mycoplasma pneumoniae* outbreak and change in reporting policy. Moreover, the number of discharged pneumonia patients continued to increase in the elderly compared to other age groups, and the severity of comorbidities in pneumonia patients was found to increase as a result. South Korean society is expected to age even further, and comorbidities cause complications and difficulties in the treatment or management of pneumonia. Therefore, chronic diseases should be managed proactively in the elderly.

**Author Contributions:** Conceptualization, K.L.; formal analysis, J.H.; funding acquisition, K.L.; project administration, J.H.; writing—original draft, K.L.; writing—review and editing, K.H., S.K. and J.H. All authors have read and agreed to the published version of the manuscript.

**Funding:** This paper was supported by Eulji University in 2021(EJRG-21-12).

**Institutional Review Board Statement:** The study protocol was approved by the Institutional Review Board of Eulji University (IRB no. EU21-020).

**Informed Consent Statement:** Not applicant.

**Data Availability Statement:** The datasets analyzed during the current study are available at the following website: http://www.kdca.go.kr/contents.es?mid=a20303010502 (accessed on 4 March 2021.)

**Acknowledgments:** The authors thank the Korean Disease Control and Prevention Agency and Eulji University.

**Conflicts of Interest:** The authors have no conflicts of interest to declare for this study.

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
