# Peer review of "Characteristics and Epidemiology of Discharged Pneumonia Patients in South Korea Using the Korean National Hospital Discharge In-Depth Injury Survey Data from 2006 to 2017"

_2036-7449, doi:10.3390/idr13030068_

Round 1

Reviewer 1 Report

Please improve the quality of figure 2. 

Reviewer 2 Report

It's a well-written article. Please address the below questions

Is this study group included the primary diagnosis of pneumonia or took into account patients admitted with other medical conditions and found to have pneumonia?

The increase in the number of patients discharged in 2017 with pneumonia can be attributed to better documentation and advancement in diagnosis and treatment guidelines?

Can the fluctuation in the number of patients discharged from hospitals with pneumonia be related to overall hospital admission in those years?

‘Hypertension can cause diabetes ‘ is misleading and not supported by references.

Did the authors look into the association between smoking and COPD in patients who died of pneumonia? Any such association will add more value to the article.

Author Response

Reviewers' Comments to Author:

Reviewer 2

Comment #2-1:
It's a well-written article. Please address the below questions.

Is this study group included the primary diagnosis of pneumonia or took into account patients admitted with other medical conditions and found to have pneumonia?

Response #2-1:

The study population in this study consisted of patients with pneumonia as the primary diagnosis in their discharge record. Those with pneumonia as a sub-diagnosis were not included in the study.  

Comment #2-2:

The increase in the number of patients discharged in 2017 with pneumonia can be attributed to better documentation and advancement in diagnosis and treatment guidelines?

Response #2-2:

The increase in the number of discharged pneumonia patients in 2017 relative to 2006 could be attributed to mycoplasma pneumonia being designated as a legal communicable disease subject to reporting, starting from 2011.  

Comment #2-3:

Can the fluctuation in the number of patients discharged from hospitals with pneumonia be related to overall hospital admission in those years?

Response #2-3:

Korean National Hospital Discharge In-depth Injury Survey samples approximately 9% of Korean patients discharged each year. Therefore, it is associated with the total number of discharged patients.  

Comment #2-4:

‘Hypertension can cause diabetes’ is misleading and not supported by references.

Response #2-4:

To clarify, we revised the sentence as below.

(Page 9) In addition to cardiovascular diseases, hypertension could increase the risk of diabetes, and this demonstrates a high level of correlation between chronic diseases.

Comment #2-5:

Did the authors look into the association between smoking and COPD in patients who died of pneumonia? Any such association will add more value to the article.

Response #2-5:

That issue has been mentioned as a limitation of the study.  

(Page 10) In addition, despite the fact that pneumonia is associated with personal health behavior, such as smoking, drinking, and living environment, these factors were not considered due to the limitation in the data used in this study. Therefore, future studies should observe the pattern of pneumonia with consideration at individual, hospital, and regional levels.  

Reviewer 3 Report

The manuscript "Characteristics and epidemiology of discharged pneumonia patients in South Korea using the Korean National Hospital Discharge In-depth Injury Survey data from 2006 to 2017 " is considered adequate to the Infectious Disease Reports.

However, is required English professional proofreaders in this paper.

The abstract is a perfect briefing of the study.

In the introduction, the approach used to the pneumonia is covering the requirements by the Journal. The methodology is comprehensive and clear regarding the objectives. 

Nonetheless, it is suggested in the body of the paper to insert a map of Korea or Cities with the representative percentage or incidence data of pneumonia.

This map will help the reader to localize in the space and observe the statistics according to the regions, regarding the data available from the last year. 

The location and data will facilitate also in a way that enables a greater understanding of the relationships between locations and the discovery of spatial patterns in the data that it is exploring.

In relation to the Discussion, the description used is adequate to the paper.

Despite that, is suggested to include a final consideration and/or recommendations for future studies around the same themes. 

Notwithstanding, the evidence presented in the paper is highly recommended to publish.
